# Lack of Habituation in Migraine Patients Based on High-Density EEG Analysis Using the Steady State of Visual Evoked Potential

**DOI:** 10.3390/e24111688

**Published:** 2022-11-18

**Authors:** Msallam Abbas Abdulhussein, Zaid Abdi Alkareem Alyasseri, Husam Jasim Mohammed, Xingwei An

**Affiliations:** 1Department of Biomedical Engineering, College of Precision Instruments and Optoelectronics Engineering, Tianjin University, Tianjin 300072, China; 2Faculty of Computer Science and Mathematics, University of Kufa, Najaf 54001, Iraq; 3ECE Department, Faculty of Engineering, University of Kufa, Najaf 54001, Iraq; 4College of Engineering, University of Warith Al-Anbiyaa, Karbala 63514, Iraq; 5Information Technology Research and Development Centre, University of Kufa, Najaf 54001, Iraq; 6Department of Business Administration, College of Administration and Financial Sciences, Imam Ja’afar Al-Sadiq University, Baghdad 10001, Iraq; 7Academy of Medical Engineering and Translational Medicine, Tianjin University, Tianjin 300072, China

**Keywords:** migraine, SSVEP, habituation, HD-EEG, entropy, cortical spreading depression

## Abstract

Migraine is a periodic disorder in which a patient experiences changes in the morphological and functional brain, leading to the abnormal processing of repeated external stimuli in the inter-ictal phase, known as the habituation deficit. This is a significant feature clinically of migraine in both two types with aura or without aura and plays an essential role in studying pathophysiological differences between these two groups. Several studies indicated that the reason for migraine aura is cortical spreading depression (CSD) but did not clarify its impact on migraine without aura and lack of habituation. In this study, 22 migraine patients (MWA, N = 13), (MWoA, N = 9), and healthy controls (HC, N = 19) were the participants. Participants were exposed to the steady state of visual evoked potentials also known as (SSVEP), which are the signals for a natural response to the visual motivation at four Hz or six Hz for 2 s followed by the inter-stimulus interval that varies between 1 and 1.5 s. The order of the temporal frequencies was randomized, and each temporal frequency was shown 100 times. We recorded from 128 customized electrode locations using high-density electroencephalography (HD-EEG) and measured amplitude and habituation for the N1–P1 and P1–N2 from the first to the sixth blocks of 100 sweep features in patients and healthy controls. Using the entropy, a decrease in amplitude and SSVEP N1-P1 habituation between the first and the sixth block appeared in both MWA and MWoA (*p* = 0.0001, Slope = −0.4643), (*p* = 0.065, Slope = 0.1483), respectively, compared to HC. For SSVEP P1–N2 between the first and sixth block, it is varied in both MWA (*p* = 0.0029, Slope = −0.3597) and MWoA (*p* = 0.027, Slope = 0.2010) compared to HC. Therefore, migraine patients appear amplitude decrease and habituation deficit but with different rates between MWA, and MWoA compared to HCs. Our findings suggest this disparity between MWoA and MWA in the lack of habituation and amplitude decrease in the inter-ictal phase has a close relationship with CSD. In light of the fact that CSD manifests during the inter-ictal phase of migraine with aura, which is when migraine seizures are most likely to occur, multiple researchers have lately reached this conclusion. This investigation led us to the conclusion that CSD during the inter-ictal phase and migraine without aura are associated. In other words, even if previous research has not demonstrated it, CSD is the main contributor to both types of migraine (those with and without aura).

## 1. Introduction

The World Health Organization (WHO) listed the top 20 global causes of disability, including migraines, which impact 50% of the population on average. Migraine is a recurrent neurological type of malfunction that is frequently accompanied by excruciating, incapacitating headaches. Migraines typically have two phases known as the interictal and peri-ictal phases, the latter of which has four stages named pre-ictal, prodrome, ictal, and post-ictal phases. These phases are also accompanied by a variety of physical and psychological symptoms, including irritability, pain, exhaustion, difficulty speaking, aversion to light and noise, vomiting, and tingling numbness [1,2]. The difference between the two main types of migraines, classic and common migraine, is the presence or absence of spreading oligemia in the case of the migraine with the aura as opposed to the migraine without the aura [3,4]. Therefore, the topic of whether these two types of migraine should be viewed as fundamentally distinct entities, or as various expressions of the same underlying pathophysiology has been a source of debate among neurologists and headache researchers [5]. The most significant migraine trait that all migraineurs share is the malfunction of the neural cortical excitabilities to any repetitive or sensory stimulation during the inter-ictal phase as the primary migraine-specific biomarker (known as habituation deficiency [6,7]. Golla and Winter noticed that SSVEP amplitudes are increased and there is a lack of habituation in the migraineurs when they are compared to the healthy control in 1959. Since then, this characteristic has become the hallmark of migraineurs, and EEG rhythmic activities have become a key factor in the diagnosis of the migraine [8]. Migraines can be brought on by a variety of factors, including environmental factors, female hormone changes, and genetic abnormalities. However, cortical spreading depression (CSD) has been shown in the animal model to be able to stimulate the peripheral and central trigemi-novascular neurons in inter-ictal changes, which may predispose them to migraine, especially with aura [9,10,11,12]. This (CSD) is stimulated by environmental factors, particularly visual ones where (SSVEP) is used for evaluating the visual response of the migraineurs and the healthy participants. These visual responses measure the activity of the neurons in the visual cortex and follow the majority of the significant change in electrophysiological activities in migraineurs’ visual cortex over these courses of the migraine cycles. Recently, numerous research teams have shown significant variations in bioelectrical activity for the visual cortex of migraineurs with or without aura over the course of migraine cycles [13]. SSVEP recordings revealed that both types of migraineurs’ brains exhibit an inter-ictal lack of habituation and considerably increased amplitude upon repeated stimulation [14,15,16,17]. The activity of visual cortex neurons, which determines the variance in habituation and amplitude between MWA and MWoA, must therefore be confirmed.

To arrive at this conclusion, we decided to use a high-density EEG to assess habituation and amplitude at SSVEP N1–P1–N2 in healthy controls (HC), migraine with aura (MWA), and the migraine without aura (MWoA). In fact, high-density electroencephalography (HD-EEG) is gaining popularity and is being used to study electrophysiological resting state functional connectivity. EEG signal source localization on the cortical surface is made possible by the higher resolution, which is provided by HD-EEG and typically has 128 or 256 channels [18]. Despite, the high cost, setup, and interpretation time for HD-EEG, which impedes widespread implementation, studies have demonstrated improved localization of the epileptogenic cortex using this HD-EEG. Furthermore, HD-EEG supports many research techniques such as functional connectivity including acquisition time, acquisition length, source localization techniques, eyes closed open, channel density, and coherence [19]. Then we processed the EEG- signal by mathlab\eeglab version 2.6, and to evaluate the statistical differences, IBMSPSS- statistical version 26 is used. ANOVA was used to determine the comparison between HC-MWA and MWOA, which contributed to significant effects besides those from the general linear model technique, and two models of the T-test were used to efficiently extract features from SSVEPs models.

Entropy is typically a measure of how accurately physiological signal complexity reflects the strength of brain systems [20]. Therefore, the use of recently developed temporal entropy analysis tools has allowed us to better understand brain dynamics and evaluate how complexity conveys information about a variety of physiological systems in addition to the measurements mentioned above. Indeed, to assess complicated signals, many entropy analysis techniques are used [21]. In this article, we concluded that (MWA) would demonstrate common and particular neurophysiological abnormalities at a higher level than (HC) and (MWoA), which conforms to the results of the findings from earlier studies about the fact that SSVEP habituation did not occur during the inter-ictal phase. Since entropy measures how accurately a physiological signal that is complex reflects the strength of brain systems, it has helped us to better understand brain dynamics and evaluate how complexity conveys information about a variety of physiological systems [22].

## 2. Related Work

Should be noted that the steady-state evoked potentials (SSVEP), used for the first time by Golla and Winter in 1959, described an increased photic response and behavior-reduced habituation when exposing the migraineurs to stimulation frequencies above 10 Hz compared to healthy controls [21]. There are three frequencies of response steady-state evoked potentials (SSVEP), low (about 10 Hz), medium (about 20 Hz), and high (about 40 Hz), and they are generated via the synchronization of cortical neurons to the stimulus frequency. The (SSVEP) is different from transient potentials many quantitative EEG studies indicated medium and high stimulation frequencies featuring spectrum presence of increased power because their constituent discrete frequency components stay closely constant in amplitude and phase over a lengthy duration [22]. The majority of research has used repeating flicker visual stimuli and shown that migraine with and without aura has increased amplitude of the spectral component (F1) in the stimulation frequency range of 15–30 Hz during the inter-ictal phase [19,23,24]. In the climacteric stage, the amplitude of component (F1) reverts to normal boundaries, supporting the unstable characteristic of migraine brain dysfunction despite there being some differences in visual stimuli processing between the two types of migraine [25]. The subtle differences between the two forms of migraine in visual stimuli processing were confirmed by Shibata et al., when they used a pattern reversal stimulus paradigm. They concluded that, in comparison to migraines without aura and controls, migraines with aura displayed an elevated amplitude and high contrast [26]. Indeed, these functional variations are in fact related to the mode of stimulation and reflect it in the form of an aura [27]. The steady-state responses vary over time in a more sophisticated way in accordance with intricate patterns of variability and do not show signs of habituation compared with averaged evoked potentials [28,29]. These important investigations into visual steady-state responses provided a preliminary description of the migraine brain in terms of abnormal neuronal network oscillations. Accepted Manuscript extraneous stimuli, especially those of the visual variety, could change the brain’s resonance to rhythms due to complex mechanisms of thalamo-cortical dysregulation. These investigations revealed that the susceptibility to external sensory stimuli was due to a baseline thalamo-cortical dysrhythmia [30]. These theories have been confirmed by subsequent works inspecting synchronization and functional connectivity techniques used to spontaneous and evoked EEG signals (Table 1).

Since CSD occurs during the inter-ictal phase, which is marked by habituation defects for migraine with aura, several researchers have recently come to the conclusion that it may be the cause of migraine seizures. In this study, we came to the conclusion that migraine without aura and CSD during the inter-ictal phase are related. In other words, despite the fact that previous studies have not shown it, CSD is the primary cause of both types of migraine (those with and without aura).

## 3. Materials and Methods 

### 3.1. Subjects

The average age (29 ± 1) of the 22 migraine patients in this study was noted. Average age (27 ± 1) of the healthy controls (HV, N = 19, 8 females), the MWA (N = 13, 7 females), and the MWA (N = 9, 5 females) at the Al-Ahram laboratory [3]. Before the experiment date, we reviewed patient records to gather information on a variety of clinical variables, including the number of migraine attacks each month, the length of the attacks, and the duration of the attacks (Hours) we chose SSVEP during the inter-ictal phase. In this stage, the migraineurs sufferers, aura or not, are distinguished by a lack of habituation to repeated stimuli as well as ictal normalization [32,33]. The key inclusion criterion in this study is the absence of attacks at least three days before and following this recording session. This is verified by gathering headache diaries and by conducting in-person or telephone interviews. In this study, the inter-ictal interval was monitored in all migraine patients at Al-Ahram Laboratory in the R.E.

To provide a baseline for the comparison, we enrolled a group of 19 healthy volunteers (10 women, mean age of 27) who were also recruited through advertisements and randomly chosen from a patient population. Each participant earned $10 (Table 2) in addition to receiving a thorough explanation of this study and receiving written informed consent.

### 3.2. Stable State Visually Evoked Potentials (SSVEP)

The SSVEP stimulation was created and shown using Psychtoolbox coupled to MATLAB [34,35]. Throughout the course of the experiment, vertical sine wave colorless gratings were introduced at (0.05 cd) presented and were superimposed on central gratings. Before SSVEP recordings, each participant adjusted with ambient room lights for 10 min before gratings alternated contrast at 4 Hz and at 6 Hz according to Equation (1) for 2 s, followed by an inter-stimulus interval which varied between the 1 and 1.5 s. This was done to obtain a steady pupillary diameter. A homogeneous brightness field of five cd/m^2^ was present around a TV monitor in this acoustically isolated room with muted lighting. At the 4000 Hz. and 35 Hz low-pass digital filter was used after collecting and sampling 600 sequential sweeps, each lasting 200 ms. The cortical response was divided into six successive blocks of 100, each of which had at least 95 sweeps that were free of artifacts. The average replies for each block were computed using Signal^TM^ software package version 4.1.
(1)$$L\left(X,Y\right)=L_{0}\left[1+C\;exp\left(-\frac{{\langle X-X\rangle }^{2}}{2\sigma_{Xs^{2}}}-\frac{{\langle Y-Y\rangle }^{2}}{2\sigma_{Ys^{2}}}\;\right)\sin\left(2\pi f_{s}\;\left(X-X_{s}\right)+\emptyset \right)\right]$$

(Figure 1A) shows a black circle in the center for fixation (two increments) for a period of 0.1–1.5 s, followed by a grey screen and a stimulus of 4–6 Hz for Steady-State Visually Evoked Potentials (SSVEP) (2 s). When the circle flashed white, the subjects would press the spacebar (for 0.1 s). If the subjects answer, the fixation circle turns red for a brief (1–1.5 s) time before turning black (for 0.1 s). Responses greater than 1s were excluded and so were those that preceded the color change.

If the signal amplitude is greater than 90% of the range of the analog-to-digital converter (ADC), the artifacts are automatically rejected using the Signal^TM^ artifact rejection tool, which is managed by visual inspection. Because background EEG amplitudes differ between participants, this method ensures that all severe artifacts are artifacts but does not systematically delete any signal. It does offline correction of liner DC drift, eye movement, and the blinks for the evoked potential signal (Figure 1B).

According to their latencies, the following SSVEP parts are recognized: N1 is recognized as the negative peak between 60 and 90 milliseconds, the P1 is the most positive peak between 80 and 120 milliseconds after N1, and N2 is the most negative peak between the 125 and 150 milliseconds after P1. The N1 to P1 and P1 to N2 complexes’ peak-to-peak amplitudes were measured. Habituation was assessed using the slopes of linear regression lines for six blocks (Table 3).

### 3.3. EEG Capture

The central occipital, parietal, and frontal regions were covered with high resolution using a specially positioned high-density EEG that had 124 electrodes with a 14-mm spacing between them situated on a nylon cap. With the use of a BioSemi Active Two system, EEG data were captured (BioSemi, Amsterdam, Netherlands). Then, 24-bit A/D conversion is used to digitize data using addigitize. Four additional electrodes were positioned around the eyes to detect the electrooculography (EOG) signals. The standard CMS and DRL electrodes from BioSemi are operated online. 

## 4. Statistics

For our analyses, we used IBM SPSS 26.0, and Microsoft Excel 2019 to create the charts. The early descriptive data for age and gender reveal a normal distribution of the v by time in each BLOCK in each group (i.e., for SSVEP N1 to P1 and P1 to N2 peak to peak); nevertheless, K, Kolmogorov–Smirnov indicates that there is a non-normal distribution between BLOCKs and groups (Table 4). For analysis, “between-factor” subject and “within-subject” factor are both susceptible to the general linear model approach (GLM). The within-subject component is “blocks”, and between the subject factors are “group” (HC vs. MWA or HC vs. MWoA).

To determine which comparison (s) contributed to significant effects, two models of *t*-test followed by an analysis of variance (ANOVA) were used. Post hoc testing was then carried out using Tukey’s Honest Significant Differences test (HSD).

We used an ANOVA for slope, with the group factor set to “group,” and Tukey tests for the post hoc analysis. Additionally, the partial eta2 and op were employed for ANOVA. The cutoff for statistical significance was (*p* < 0.04) SSVEP amplitude slopes and clinical variables were examined for relationships using Pearson’s correlation test.

## 5. Results

When a *t*-test is used to compare the times in N1, P1, and N2, it is observed that while there are no significant variations between normal and patient times in N1, there are significant disparities for both P1 and N2 times. N1, P1, and N2 between BLOCKs and have no significant differences in time, however, there are subs, potential variances in time at the level of significance (0.04). In MWA, P1 advances in both BLOCKs (1, 2) before HC despite a brief delay, whereas P1 looks to be much behind HC in BLOCKs (3, 6). Regarding MWoA, P1 in each of BLOCKs (1,3,5) has a considerable and meaningful delay, but BLOCKs (2,4,6) have an early statistically significant delay (Figure 2).

Entropy analysis reveals considerable variations between HC and migraine patients of the main kinds, resulting in important distinctions. There are no substantial differences in time between these differences on P1–N2 times or N1–P1 with N2 amongst BLOCKs, but there are minor, potentially significant variations in time. Despite a slight delay, P1 advances in both BLOCKs (1, 2) before HC in MWA, although P1 appears to be much behind HC in BLOCKs (3, 6). Regarding MWoA, P1 has a significant and meaningful delay in each of the BLOCKs (1,3,5), whereas BLOCKs (2,4,6) have an early statistically significant delay. Moreover, it can be seen that MWA and MWoA exhibit a decrease in amplitude as well when we use the entropy (Figure 3).

Our results indicate that CSD is closely related to the difference between MWoA and MWA in the absence of habituation and amplitude decrease in the inter-ictal period. Numerous researchers have recently come to this conclusion because CSD appears during the inter-ictal phase of migraine with aura, which is when migraine seizures are most likely to happen. Our analysis led us to the conclusion that migraine without aura and CSD during the inter-ictal phase are related. In other words, even if earlier studies have not supported it, CSD is the primary cause of both main types of migraine.

By comparing the graph (Figure 2) and the highest and lowest values, it was discovered that N1, P1, and N2 differed in time, indicating that the most notable difference between the highest and lowest value (N1–P1 and P1–N1) occurred at various times in various BLOCKs and groups.

Even if P1 time arrives earlier and remains obvious, there is a distinct delay in the BLOCKS for N2 in MWA that is smaller than in the case of MWoA. Therefore, based on the change or shift in the normal state with a delay of N2, the case can be predicted, known, or diagnosed using an electroencephalogram. It may be assumed that the individual is ill if the delay exceeds 10% (Figure 4).

## 6. According to an ANOVA Analysis, There Were Significant Differences between Groups in All Voltage Differences (N1–P1 and P1–N2)

The differences between HC and MWA in N1-P1 were greater than those between HC and MWoA, although the differences between MWA and MWoA are more important than those between MWOA and HC. The best for diagnosis is consequently MWoA, where the difference between the healthy case and the case with aura increases less and the gap between them substantially increases. It is preferable to analyze the two relationships of length or time with this instance since the pathological factors affecting MWA and MWoA differ more between autistic individuals and healthy cases. There were no differences between MWoA and HC in P1–N2, although there were maximal disparities between HC and MWA and minimal differences between MWA and MWoA. The N1, N2, and P1 points must be determined by studying the amplitude. We use the difference N1-P1 to identify the case types, that is, whether this disease is MWA or MWoA, and the time shift to confirm or deny the presence of a disease. The distinctions between P1 and N2 should not be used because they can produce unclear results (Figure 5).

## 7. Discussion

The purpose of this conducted study is to display the electrical investigation of the visual cortical response in migraine by the search for differences between the two distinct phenotypes of migraine without aura (MWoA) and migraine with aura (MWA). Firstly, we confirm earlier findings that the P1–N2 component of the SSVEP will never habituate over the subsequent block of the averaged response in migraine patients with the aura but does in healthy controls (HC). This situation also occurs for SSVEP N1–P1, and P1–N2 components [33,36]. Additionally, a noteworthy discovery is that, as compared to HC, lack of habituation (SSVEP N1–P1) in MWA more than in MWoA. 

Secondly, a noteworthy finding is that the magnitude of visual responses varies between MWoA and MWA. The N1–P1 VEP amplitudes are consistently higher in MWA patients than those in MWoA. In contrast to MWA, MWoA patients exhibit similar SSVEP N1–P1 and P1–N2 block amplitudes to HC but have lower habituation SSVEP responses during the six successive blocks of 100 averages. 

As far as we are aware, this is the first study to use ultra-high-density EEG to distinguish two classes of migraine (MWA and (MWoA)) based on increased SSVEP N1 to P1–N2 amplitudes and decreased habituations. In our work, the average responses of the SSVEP N1 to P1 and P1 to N2 amplitudes are shown to be larger in MWA patients than in controls or MWoA patients, according to earlier VEP investigations in the groups of the migraine with the aura (MWA) patient [26,37]. In contrast, it was found in earlier research that the VEP amplitude was less in the MWA [38], even when it is compared to the MWoA [39]. While most of the time, the VEP amplitude in the MWA was noticed to be within the usual ranges [15,20]. Therefore, our findings supported earlier research that showed migraineurs had an overactive visual cortex during the inter-ictal phase [36,40]. Increased SSVEP amplitude and a lack of habituation are the results of this dysfunction, which are more pronounced in MWA than in HC or MWoA.

From a pathophysiological significance of view, it is good to compare MWA and chronic migraine which is likewise thought to be linked with genuine cortical hyperexcitability, which is confirmed by studies of magnetoencephalographic visual evoked responses and somatosensory evoked potentials [7,41]. In MWA, SSVEP amplitude was increased in virtually all blocks of averages, and habituation was deficient over six blocks, whereas in chronic migraine, only the first block of averaged visual or somatosensory responses was increased in amplitude, but not the subsequent blocks, and habituation was normal. As a result, the electrophysiological pattern in migraine with auras might indicate that the visual cortex is permanently hyperexcitable.

There are still no recognized pathophysiological explanations for the different aura forms and variations in inter-ictal visual evoked potential patterns. However, in the corresponding brain imaging investigations done on MWA during the inter-ictal phase, the concomitant vascular and metabolic abnormalities spread more widely than when the patient had no aura or visual problems [42,43]. As a result, the (MWA) is a cortical spreading depression (CSD) that is caused by an electro-chemical wave that typically originates in the posterior lobe of the brain and it spreads anteriorly at a rate of around 3 mm/min, and it is accompanied by the biphasic alterations in the cerebral blood flows [44,45]. This shows that the Na^+^/K^+^ ATPase pump and intact neurovascular play a significant role in the recovery from CSD by coordinating increased energy demands and restoring the ions gradients. Additionally, the extent to which CSD spreads via MWA seizures emphasizes an aura clinical pattern.

While the neurovascular tones are influenced by local factors, such as oxygen availability, lactate concentration, and subcortical monoaminergic inputs [36,46], in migraine sufferers particularly those with aura, continuous visual stimulation disrupts neurovascular coupling in inter-ictal [47,48,49]. Additional data from the biochemical and functional neuroimaging study support the hypothesis that migraine disrupts the monoaminergic transmissions from the brainstem to the thalamus and cortex [50]. Convergent evidence from numerous laboratories also suggests that migraineurs’ brains’ levels of ATP and mitochondrial energy reserve are significantly reduced in between attacks [51,52]. 

Numerous studies have also found an inverse relationship between migraine auras and the rate of phosphocreatine (PCr)/phosphate (Pi), a marker of the brain’s energy reserve. The complexity of the aura increases with decreasing rapidity [30,53]. Additionally, 1H-MR spectroscopy showed that during prolonged visual stimulation in patients with visual auras, lactate levels in the visual cortex rose. In contrast, HV and MWoA patients did not experience this [54]. Epigenetic studies on migraine have connected, hormonal fluctuations to changes in DNA methylation and gene expression, which may be a factor in the metabolic differences amongst aura forms. Even though numerous exploratory studies have discovered numerous genes and pathways that may contribute to migraines, additional in-depth genomic and functional studies to better understand processes may help with better outcomes for diagnosis and treatment [55,56]. As a result, the genetic load may affect the severity and patterns of CSD, for instance, familial hemiplegic migraine (FHM1,2), a rare form of migraine with aura, was found to have a mutation in CACNA1A, ATP1A2, and the SCN1A gene, respectively [57,58]. In Ref. [59], an experiment on mice presented that the clinical phenotype was more severe in (FHM1), indicating that the S218L mutation makes CSD more prevalent and more widespread, also there was a higher threshold for cortical spreading depression in (FHM2) as a result of an E700K mutation. However, mutations in (FHM1,2) only cause common variations in a few loci found through genome-wide association studies (GWAS) that resemble those seen in migraine without aura, rather than being the direct cause of migraine with aura, this means that the subject of their research was, how closely linked genetic mutations on mitochondrial DNA variants may impact the clinical migraine phenomenology, particularly the aura [60,61]. 

The putative relationship between the VEP anomalies identified inter-ictal and ictal events, such as CSD, and its spread can only be conjectured. According to Siniatchkin, greater VEP amplitudes were associated with the extension of the paroxysmal EEG activated toward more anterior brain regions in the photosensitive individuals with photo paroxysmal responses to the intermittent photic stimulations. These electrophysiological correlates with the increased functional connectivity between the occipital and parietal-temporal-frontal networks, which are under the control of the thalamus may be seen in phototactically generated seizures and photo-paroxysmal responses [36,62]. According to a recent study, CSD can cause disturbance in the transmission of sensory information to the brain, which helps to explain several characteristics of migraine with aura during the inter-ictal phase [63,64]. This pathophysiological mechanism underpinning the VEP habituation is not permanently impacted by ictal events, regardless of the link between the ictal CSD and the inter-ictal VEP. These MWA groups are more pronounced pathophysiological dysfunctions as a result of genetic abnormalities that produce cortical spreading depression, which fuels meningeal nociception in migraine with aura, and are particularly affected by the lack of habituation between inter-ictal [16,65,66]. This shows that when the interval between migraine attacks lengthens, inhibitory performances and the habituations with the stimulus recurrence decline. This is supported by our current data. An association between the inhibitory process and number of the days since the previous attacks were found in psychophysical studies that applied the visual masking test [67,68]. For the magnocellular system, the lower spatial frequencies are preferred, which is thought to play a major part in processing transitory visual inputs. This may help to explain why people with migraines are more adept at identifying quickly presented stimuli [67]. Since visual stimulus downregulates GABAergic neurotransmitters in a concentration-dependent manner, lactate levels in the occipital brain of MWA are elevated in response to the visual stimulations. Maybe because of the biochemical correlation of impaired inhibitory mechanisms that could be lactate-induced downregulation of GABA activity in the occipital cortex [69], greater GABA levels are linked with a higher migraine burden [70].

Our methodology has a few flaws, similar to other studies on neurophysiology [36,71]. For instance, investigators were blinded during the offline analysis of the SSVEP data, as they were in earlier investigations by the separate groups, but they were not during the diagnosis and recording session. In fact, in clinical settings, it is very challenging to completely blind someone. Additionally, we are aware that clinical correlations are retrospective and that our sample sizes are constrained. Therefore, a longitudinal, and prospective follow-up of the patient is required in future studies to duplicate the findings in larger clinical samples with a variety of migraine morphologies and to record patients both during attacks and at various times between episodes.

## 8. Conclusions and Future Work

This study confirms the findings of previous studies, which said that migraine patients in the inter-ictal phase suffer during continuous sensory stimulation from increases in amplitude and habituation deficit. Despite this, migraineurs with aura are different from those without auras, these differences are represented by the time of defect in habituation and its value. As a result, in both kinds, migraine occurs due to cortical spreading depression (CSD). The disparity between the two types of migraine is a result of the difference in the pathway that CSD takes in the brain in spite of the CSD pathway, which may be caused by a genetic mutation that remains to be determined. Numerous experts have lately concluded that CSD may be the cause of migraine seizures since it happens during the inter-ictal phase of migraine with aura. This investigation led us to the conclusion that there is a connection between migraine without aura and CSD during the inter-ictal phase. In other words, despite the lack of evidence from earlier studies, CSD is the major contributor to both types of migraine (those with and without aura). Therefore, to demonstrate the amount and rate of diffusion for CSD, the researchers had to assess coherence via the entropy technique for both forms of migraine during the inter-ictal phase. Preventive drugs for migraine patients, such as topiramate therapy, enhance the cortical processing of sensorial stimuli and follow an improvement in habituation in migraineurs. Thus, finding out how this medication affects both types of migraines is a future objective.

## Figures and Tables

**Figure 1 entropy-24-01688-f001:**
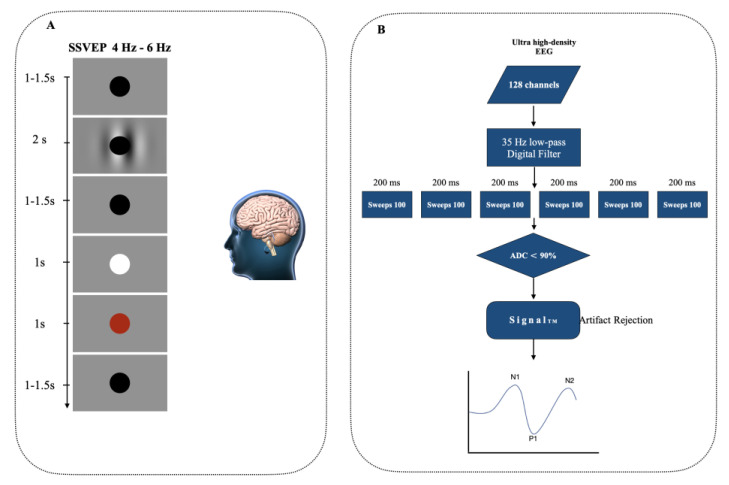
(**A**). Circle centrally for fixation (2 inc) to a period of 1–1.5 s. followed by a stimulus of 4–6 Hz for Steady-State Visually Evoked Potentials (SSVEP) consisting of a grey screen and a fixation black circle for a period (2 s). The subjects pressed the space key whenever the circle flashed white (for 0.1 s). If subjects respond the circle becomes black for a period (1–1.5 s), if they did not, the fixation circle turned red (for 0.1 s). (**B**) SSVEP preprocessing, 600 sequential sweeps divided into six successive blocks each lasting (200 ms).

**Figure 2 entropy-24-01688-f002:**
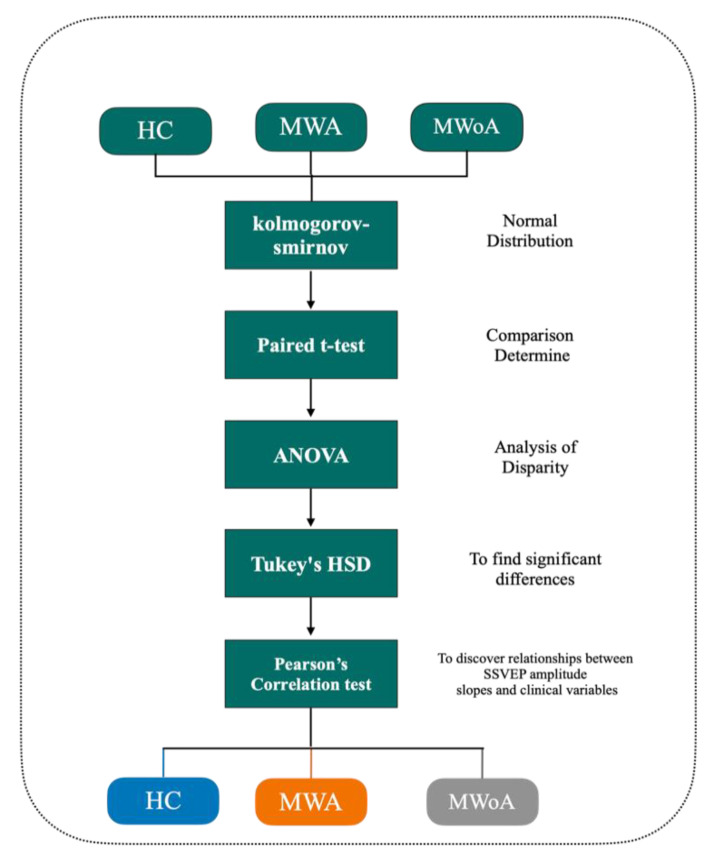
We use paired T-test followed by ANOVA analysis of variance between HC and MWA, HC and MWoA.

**Figure 3 entropy-24-01688-f003:**
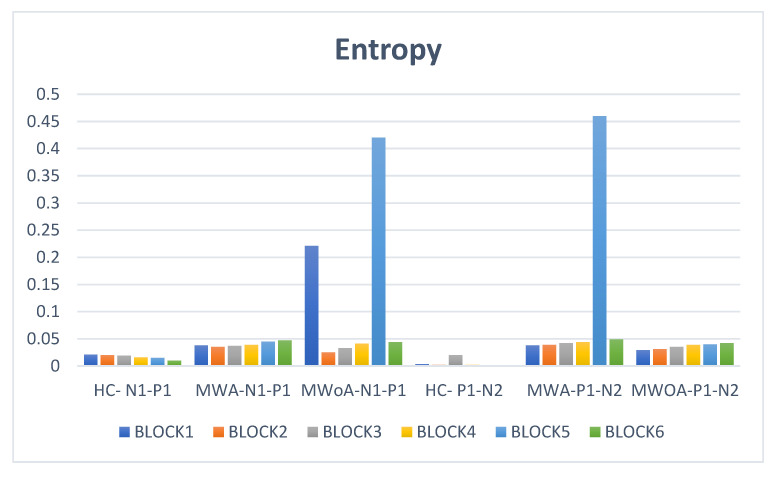
Entropy analysis represents the variations between HC and migraine patients with MWA and MWoA.

**Figure 4 entropy-24-01688-f004:**
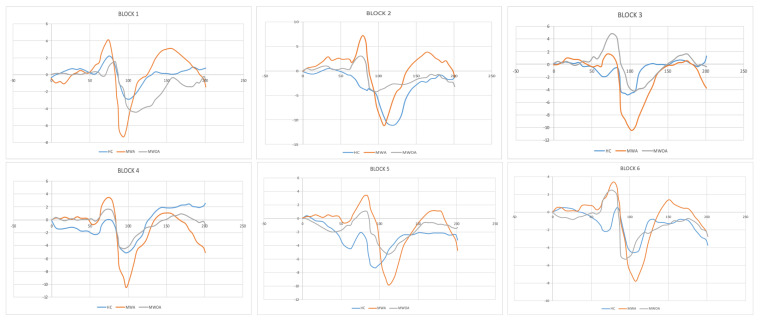
Representative recordings (low pass filter 35 Hz) of SSVEP, Healthy Control (HC), migraine patient without aura (MWoA), and a migraine patient with aura (MWA) in the interictal phase. We compared every three blocks together at 100 averaged responses to illustrate the difference between block N1–P1 and P1–N2 amplitudes and amplitude change (habituation) through six blocks.

**Figure 5 entropy-24-01688-f005:**
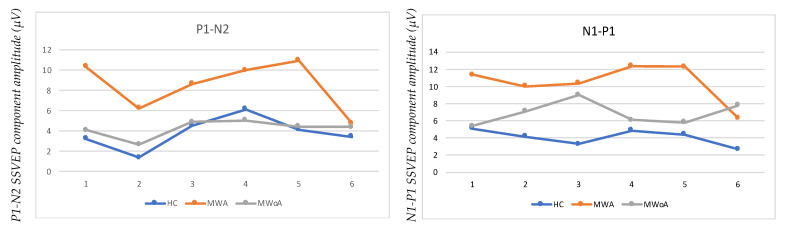
SSVEP component in six sequential blocks of the 100 recordings to compare with the healthy controls (HC), migraine with aura (MWA), and migraine without aura (MWoA) N1–P1–N2.

**Table 1 entropy-24-01688-t001:** Summarize previous works.

	Authors	Method	Cases	Phase	Result
1	(Golla and Winter 1959) [21]	3.5–25	50 MIG 113 HC	Inter-ictal	Increased amplitude and lack of habituation (SSVEPs)
2	(Simon et al., 1982) [23]	10–40 Hz	11 MIG 11 HC	Inter-ictal	Increased power in the alpha band at 24, 34, 38 Hz stimulation
3	(Nyrke, Kangasniemi, and Lang 1989) [25]	10–24 Hz	30 MWA 20 MWoA 49 HC	Inter-ictal	In MWA, increased power at 16–22 Hz, in MWoA reduced 2nd harmonic
4	(Genco et al., 1994) [24]	15–30 HZ	15 teen MWA (10–18) 25 teen MWoA (10–17) 11 teen HC (10–18) 20 adult MWA (19–45) 22 adult MWoA (19–45) 20 adult HC (18–45)	Inter-ictal	Increased amplitude of the SSVEP in all migraine patients
5	(de Tommaso et al., 1998) [27]	27 Hz	16 MWoA 20 HC	Ictal, Inter-ictal	In inter-ictal phase increased power in MWoA, In ictal phase normal power in MWoA and HC
6	(de Tommaso et al., 2003) [28]	3 Hz, 6 Hz, 9 Hz	15 MWoA 15 HC	Inter-ictal	No habituation phenome in MWoA and HC, increased power at 3 Hz in MWoA
7	(Shibata et al., 2008) [26]	5–10 Hz	10 MWoA 10 MWA 20 HC	Inter-ictal	MWA and MWoA have abnormal excitability in the primary visual cortex and significantly increased amplitude to SSVEP
8	(Shibata et al., 2011) [29]	0.5, 1.0, 2.0, 4.0 (cpd) with a stimulus rate of 7.5 Hz	12 MWoA 12 MWA 12 HC	Inter-ictal	MWA and MWoA showed high amplitude to SSVEPs and did not reveal a clear lack of habituation
9	(Fogang et al., 2015) [31]	5 Hz, 10 Hz, 15 Hz, 20 Hz	171 MWoA 61 MWA 48 C M 24 HV	Inter-ictal	The lack of habituation of cortical responses during repetitive stimulation might identify subgroups of migraine patients on spectral analysis of the EEG because of normal habituation in chronic migraine of the evoked activities and PD lower power.

**Table 2 entropy-24-01688-t002:** Clinical and demographic characteristics of the healthy volunteer (HV), and total groups of the migraine with aura patients (MWA), and migraine without aura (MWoA). The data are presented as mean SD.

Demographic Data and Clinical Characteristics	HC (*n* = 27)	MWA (*n* = 13)	MWoA (*n* = 19)
Women	8	7	5
Age (years)	27 ± 1	28 ± 1	30 ± 1
Duration of migraine history (years)		15.2 ± 8.1	14.1 ± 2.3
Attack frequency/month (n)		2.8 ± 2.1	2.1 ± 1.2
Attack duration (hours)		28.8 ± 19.7	24.6 ± 20.3
Days since the last migraine attack		14.7 ± 18.1	19.9 ± 17.1

**Table 3 entropy-24-01688-t003:** The Latencies in the milliseconds of SSVEP in the healthy control (HC), migraine patient without aura (MWoA), and migraine patients with aura (MWA) and the visual aura. Data are expressed as means ± SD.

EEG Parameters (ms)	HC	MWoA	MWA
N1 (75)	74.7 ± 1.2	74.7 ± 6.3	74.7 ± 5.5
P1 (100)	102.1 ± 0.1	101.7 ± 7.4	101.1 ± 3.2
N2 (145)	135.4 ± 3.2	156.8 ± 6.5	150.3 ± 7.3

**Table 4 entropy-24-01688-t004:** The N1–P1–N2 SSVEP components amplitude and habituation slopes in the healthy control (HC), migraine patients without aura (MWoA), and migraine patients with the aura (MWA).

	HC-N1–P1	MWA-N1–P1	MWoA-N1–P1	HC-P1–N2	MWA-P1–N2	MWOA-P1–N2
BLOCK1	5.0942227	11.3879	5.426988	3.2195488	10.34833	4.062311
BLOCK2	4.2105263	10	7.122642	1.3815789	6.231884	2.641509
BLOCK3	3.3333333	10.38043	9.027778	4.5075758	8.641304	4.861111
BLOCK4	4.9017821	12.35472	6.097765	6.1074421	9.960602	5.011181
BLOCK5	4.4410654	12.27959	5.841923	4.1069709	10.91944	4.440718
BLOCK6	2.7193086	6.375129	7.819883	3.4170887	4.753607	4.360018

## Data Availability

Data supporting by Al-Ahram Laboratory in the R.E.

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
