# Peer review of "Lack of Habituation in Migraine Patients Based on High-Density EEG Analysis Using the Steady State of Visual Evoked Potential"

_entropy, 2022, doi:10.3390/e24111688_

Round 1

Reviewer 1 Report

The introduction shoul be re-written; at present, it is very confuse. 

Methods section should be improved, and better organized. 

Results should clearly show what was planned in methods. This reviewers did not find the value of enthropy mentioned in methods. Statistics also shoudl be improved and reviewed by a stastistician. 

Discussion lack strenght not having clearly orientation with the aims apparently proposed. 

English language should be improved and reviewed by a native English speaker. 

Author Response

Thank you for your comments, we have addressed all your comments in the attached file. 

Reviewer 2 Report

most of the patients with migraine normally use medications. there is no analyses of this drugs. this should be added and additionally used as confounder in a next statistical analysis.

Author Response

Thank you for your comments, please find in the attached file our response to your comments.

Round 2

Reviewer 1 Report

Grammar corrections, albeit minor, are recommended. 

Reviewer 2 Report

please check again in the sample the use of medications and calculate the confounding impact.